# Challenges for Nanoscale CMOS Logic Based on Two-Dimensional Materials

**DOI:** 10.3390/nano12203548

**Published:** 2022-10-11

**Authors:** Theresia Knobloch, Siegfried Selberherr, Tibor Grasser

**Affiliations:** Institute for Microelectronics, TU Wien, Gußhausstraße 27–29/E360, 1040 Vienna, Austria

**Keywords:** 2D materials, field effect transistors, CMOS logic, nanoscale devices, nanosheet FET, contact resistances, Schottky barriers, van der Waals interfaces, charge traps, process integration

## Abstract

For ultra-scaled technology nodes at channel lengths below 12 nm, two-dimensional (2D) materials are a potential replacement for silicon since even atomically thin 2D semiconductors can maintain sizable mobilities and provide enhanced gate control in a stacked channel nanosheet transistor geometry. While theoretical projections and available experimental prototypes indicate great potential for 2D field effect transistors (FETs), several major challenges must be solved to realize CMOS logic circuits based on 2D materials at the wafer scale. This review discusses the most critical issues and benchmarks against the targets outlined for the 0.7 nm node in the International Roadmap for Devices and Systems scheduled for 2034. These issues are grouped into four areas; device scaling, the formation of low-resistive contacts to 2D semiconductors, gate stack design, and wafer-scale process integration. Here, we summarize recent developments in these areas and identify the most important future research questions which will have to be solved to allow for industrial adaptation of the 2D technology.

## 1. Introduction

In recent years, it has become increasingly difficult to sustain the continued downscaling in silicon technology. As large manufacturers like TSMC have repeatedly delayed mass production at the 3 nm node from late 2021 to the second half of 2022 and now to early 2023, the continued scaling results in exploding costs and extremely complex processing with more than 1000 processing steps for a single wafer. Even more, at ultimately scaled gate lengths ($L_{G}$) of below 12 nm, the scaling meets a physical limit as the channel thickness (*t*) has to be considerably reduced to maintain the required device performance. As a rule of thumb, $t=L_{G}/4$ must hold to ensure good gate control. However, for silicon, as well as any other three-dimensional (3D) semiconductor, the mobility substantially degrades at a layer thickness below 3 nm due to excessive charge carrier scattering at the interfaces [1]. As two-dimensional (2D) semiconductors are inherently only one atomic layer thick, they can provide sizable carrier mobilities when used as the channel material for scaled field-effect transistors (FETs) [2,3]. Therefore, the present version of the International Roadmap for Devices and Systems (IRDS) lists 2D materials as a component for complementary beyond-CMOS devices starting from the 1.5 nm node ($L_{G}=12$ nm) scheduled for 2028 and as a channel material for standard CMOS technology from the 0.7 nm node ($L_{G}=12$ nm) scheduled for 2034 [4]. In Figure 1a the layout of a planar Si MOSFET is compared to the layout of a double-gated 2D FET in Figure 1b.

Applications as complementary beyond-CMOS devices include a large variety of application scenarios where 2D FETs could be integrated together with silicon CMOS chips in a monolithic way at the back end of the line (BEOL). For example, a graphene-based image sensor array has been realized on a CMOS chip [5]. In this way, also photonic and optoelectronic devices like 2D material based single photon emitters [6,7], neuromorphic elements like memristors [8,9] or gas and biosensors [10,11] could be integrated on top of silicon CMOS chips. In addition, the unique physics of atomically thin semiconductors allows for new opportunities for high-performance devices at the front end of the line (FEOL). As an example, layered materials offer the possibility to form high-quality van der Waals interfaces between various materials [12,13]. This stacking of various 2D materials allows to form van der Waals heterostructures for novel device designs [14], among them designs which aim to overcome the thermal limitations of the subthreshold slope in conventional FETs, for example tunnel FETs [15] and Dirac source FETs [16]. In addition, it was recently discovered that the bandgap of layered materials can be quenched in large gate fields [17]. Even though this effect has up to now only been observable in ionic liquid gated devices, it offers new design possibilities.

In general, there are over 100 layered semiconducting compounds which could potentially serve as a channel in ultra-scaled FETs [18,19]. However, due to the practical considerations of good material availability and ambient stability as well as the desired material properties of a moderate bandgap and sizable charge carrier mobilities, most research on 2D FETs uses transition metal dichalcogenides (TMDs) as channel material. TMDs have the generalized formula MX${}_{2}$, where M is a transition metal of the groups 4–10 and X is a chalcogen. Most commonly, these materials are encountered in the metallic, trigonal 1T, and the semiconducting, hexagonal 2H phase [20], see Figure 1b. In the following, the materials are studied in their hexagonal phase with the symmetry group P-6m2 in the monolayer form. Specifically, the properties of MoS${}_{2}$, MoSe${}_{2}$, WS${}_{2}$, WSe${}_{2}$, HfS${}_{2}$, HfSe${}_{2}$, and ZrSe${}_{2}$ are evaluated, when used as a channel material for 2D FETs. In addition to these TMDs, two other 2D semiconductors are of interest as channel material, due to their particularly high charge carrier mobilities, namely black phosphorus, BP [21], and Se-terminated bismuth oxychalcogenide, Bi${}_{2}$O${}_{2}$Se [22]. BP is a layered semiconductor with an orthorhombic structure with characteristic ridges in the layer, which give rise to highly anisotropic transport, with the highest carrier mobility in the direction perpendicular to the ridges. However, BP oxidizes quickly when exposed to air [23]. In contrast, Bi${}_{2}$O${}_{2}$Se is stable for many months under atmospheric conditions and is a highly symmetric tetragonal system. In addition, Bi${}_{2}$O${}_{2}$Se is a so-called “zipper” 2D material, meaning that upon cleavage, a half filled Se layer forms its surface [22].

All in all, the theoretical projections [24,25] as well as the available prototypes [26,27] indicate a considerable potential for 2D FETs. However, the numerous challenges that still need to be solved for nanoscaled complementary metal oxide semiconductor (CMOS) logic circuits, render it unclear when and if 2D FETs will be incorporated in future technology generations [28]. In this Review, we discuss the most important problems that 2D FETs are currently facing and provide an overview about previously suggested solutions and their limitations as well as pointing towards the most important future research questions. First, the FET device design for dimensions of a few nanometers and stacked sheet device geometries need to be developed (Section 2), see Figure 1c. In terms of transistor performance, the two most essential requirements are the formation of contacts to 2D semiconductors with a small contact resistance (Section 3) and the identification of scalable gate stacks, which simultaneously allow for electrically stable device operation (Section 4). Finally, the entire fabrication process has to allow for CMOS integration of p-type and n-type FETs via doping and must be compatible with wafer-scale processing (Section 5).

## 2. Device Scaling

For the 0.7 nm node in 2034, a physical gate length of 12 nm and a gate pitch of about 40 nm is projected in the IRDS [4]. In this context, the gate pitch describes the minimum possible distance between two adjacent devices’ gates. For these ultra-scaled designs, 2D FETs are proposed to be included as a channel in an architecture of stacked channel nanosheet FETs [13,29,30]. In fact, a first prototype of a FET using two molybdenum disulfide (MoS${}_{2}$) flakes as stacked channels on top of each other has recently been demonstrated, even though the channel lengths amounted to about 400 nm [31]. While first experimental demonstrations have come close to or undercut a gate length of 12 nm [26,32] a gate pitch smaller than 40 nm remains elusive. This gate pitch describes the overall device dimensions, including access regions and contact lengths. The first promising prototypes of scaled devices with an overall pitch of 42 nm in a back-gated device design have recently been realized [33]. The following presents an overview of nanoscaled 2D FET prototypes and their respective shortcomings.

### 2.1. Ultrascaled Prototypes

Up to now, the smallest gate length ever reported for a 2D FET amounts to 0.34 nm, which was achieved by using the edge of a graphene layer as the gate electrode [32]. In this configuration, a vertical MoS${}_{2}$ channel is placed on the side wall of an etched stack, where an embedded graphene layer forms the gate. In another work, an ultra-scaled prototype was realized using a metallic single-walled carbon nanotube (SWCNT) as a back gate electrode for an MoS${}_{2}$ FET [34]. Figure 2a shows the top view of the FET using the SWCNT as a back gate and the MoS${}_{2}$ flake as a channel, and Figure 2b shows the cross-sectional transmission electron microscopy (TEM) image. While these prototypes prove that there is no fundamental limit to the gate length ($L_{G}$) of FETs down to 1 nm and achieve an impressive subthreshold slope of 65 mV/dec., see Figure 2c, the process is not scalable. The metallic CNT had to be characterized and manually aligned for device fabrication. In addition, it should be noted that the FETs had 100 nm long access regions ($L_{A}$) and that the contact lengths ($L_{C}$) and device widths (*W*) at about 2 μm were high. In a similar approach, Co${}_{2}$Si nanowires were used as a top gate for monolayer MoS${}_{2}$ FETs, where the overall device length ($L=L_{G}+2L_{A}$) amounted to only 60 nm, but the random location of nanowires again renders the approach not scalable [35]. Xu et al. have demonstrated device lengths of about 9 nm based on corrosion cracks formed in bismuth trioxide. This process of patterning nanogaps and short channel lengths is, in principle, scalable. However, it suffers from limited controllability of the device pitch due to the crack formation and large overlap capacitances between source and drain electrodes at the back and the top gate contact [36].

### 2.2. Nanoscale Transistors with Scalable Fabrication Methods

English et al. solved many of the scalability issues mentioned above in a purely top-gated design. A narrow Al stripe serves as a gate, using an Al${}_{2}$O${}_{3}$ layer formed via oxidation in air as a gate insulator and self-aligned source and drain contacts, resulting in $L_{G}=10$ nm and $L_{A}=10$ nm [26]. However, in this work $L_{C}$ at 300 nm and *W* at 1 μm are high. In contrast, the back-gated MoS${}_{2}$/SiO${}_{2}$ FETs use electron beam (e-beam) lithography in combination with reactive ion etching (RIE) to define a narrow width of 65 nm and a small device length of 50 nm. In this way, nanoscaled channel areas are created, which allow for the characterization of single charge trapping events [38]. However, due to the narrow device width, defects introduced at the edges of the etched ribbons degrade device performance, and $L_{C}$ at over 500 nm is high. A p-type tungsten disulfide (WS${}_{2}$) FET using SiO${}_{2}$ as a gate insulator with a length of 40 nm and a width of 50 nm has been realized by sulfurization of tungsten source and drain contacts, patterned close to each other to form a continuous WS${}_{2}$ film, thereby avoiding the etching process [39].

Smets et al. also used the combination of e-beam lithography and RIE to fabricate small back-gated MoS${}_{2}$/HfO${}_{2}$ FETs with device lengths down to 30 nm and $L_{C}$ of only 13 nm, which was the first prototype coming close to the targeted pitch of 40 nm [33]. Other literature reports based on e-beam lithography often achieve small device lengths of 20 nm [40], 25 nm [41] or 40 nm [42], but fail to scale the device width and the pitch. Another method for fabricating devices with scaled gate lengths of 10 nm [43] or 50 nm [44] in a back-gated device design is to use the shadow effect of pre-deposited metal electrodes for reducing the device length. In general, it should be noted that there are only few nanoscaled devices with top-gated [26,35,36] or double-gated device design [37,42], while most demonstrations of nanoscaled 2D FETs use a back-gated layout [33,34,38,39,40,41,43,44]. This discrepancy is related to one of the main difficulties faced when fabricating ultra-scaled 2D FETs, namely, the smooth van der Waals surface of a 2D material makes nucleation of the first layer in an atomic layer deposition (ALD) process difficult. Thus, the deposition of an insulating layer on top of a 2D material, essential for top-gate or double-gate formation, is a considerable challenge for the fabrication of ultra-scaled 2D FETs.

In a recent work at IMEC, excellent gate control was achieved in a connected dual gate design for extremely scaled gate lengths down to 2 nm [37]. In this design, a few layer WS${}_{2}$ channel is gated from the top and via a local back gate, where a HfO${}_{2}$ layer serves as a gate dielectric, see the cross-sectional TEM in Figure 2d. Here, a local back gate is connected through a highly n-doped well with a top contact, linking it to the top gate. In the connected dual gate configuration, short channel effects are negligible down to equivalent top gate lengths of 13 nm, see Figure 2e. Even though this design shows superior electrostatics, its gate pitch exceeds several hundred nanometers. Thus, up to now, double-gated FETs with scaled device widths, contact lengths, and gate pitches have not been demonstrated.

Another issue which is critical for scaled 2D FETs is the self-heating of the channel during device operation. In ultra-scaled CMOS logic the heat dissipation of the circuit is a limiting factor for the device performance. In addition, this problem will likely be exacerbated by a potential vertical integration of 2D based circuits in the BEOL. Furthermore, self heating is much more pronounced in 2D based FETs compared to silicon based FETs [45] and the thermal boundary conductance of the SiO${}_{2}$-Si interface is an order of magnitude higher than that of the SiO${}_{2}$-MoS${}_{2}$ interface [46]. These considerations indicate that self-heating will likely be a critical problem for nanoscaled 2D CMOS circuits.

In general, the main difficulties in the fabrication of ultra-scaled 2D FETs are the combination of sophisticated lithography methods, such as extreme ultraviolet lithography on an industrial scale or well calibrated electron-beam lithography in a university lab setting, with the challenges arising from the introduction of a novel material in the process flow. These challenges are mainly the formation of low-resistive contacts with small dimensions and the identification of an appropriate gate stack which forms a high-quality interface and allows for top-gate integration. In addition, good doping schemes are needed, allowing to process n-type and p-type FETs next to each other, enabling CMOS integration. These questions are addressed in the following sections.

## 3. Contact Engineering

In the IRDS the limit for the total parasitic series resistance ($R_{\mathrm{SD}}$) is 221 Ω
μm at the 0.7 nm node [4]. This parasitic series resistance consists of the contact resistance ($R_{C}$) and the access resistance ($R_{A}$), resulting in $R_{\mathrm{SD}}=2R_{C}+2R_{A}$, with contributions from source and drain. Therefore, even in the most optimistic case of negligible access resistances, the contact resistance must stay below 111 Ω
μm to meet the IRDS target values.

### 3.1. Impact of Schottky Barriers

One fundamental challenge for achieving small contact resistances to 2D semiconductors is the formation of Schottky barriers (SBs) at the metal to semiconductor interface. While in silicon technology Schottky contacts are generally avoided by using highly doped n or p contact regions which form Ohmic contacts with silicides, Schottky contacts are prevalent in 2D FETs [47]. This prevalence is mostly caused by the overall lack of stable doping schemes for 2D materials [48], as well as by the pronounced Fermi level pinning at the metal to semiconductor interface [49,50].

In order to decrease the contact resistance, the SB has to be reduced. In principle, small work function metals such as scandium should lead to small SBs when contacting n-type FETs based on MoS${}_{2}$ [51]. However, due to the strong Fermi level pinning resulting from an abundance of defects at the metal/semiconductor interface, the metal work function has only a small impact, creating moderate SBs. It should be noted that the direct evaporation of metal contacts on top of 2D semiconductors damages the 2D monolayer, which strongly contributes to Fermi level pinning. In fact, covalent bonds between the metal contact and the topmost semiconductor layer form [52]. Therefore, the pinning can be reduced by using transferred contacts [53], even though $R_{C}$ of the transferred contacts is high at 4000 Ω
μm.

Another possible strategy for Fermi level depinning is based on inserting an ultra-thin insulating layer between the metal and the 2D semiconductor, as was demonstrated for hexagonal boron nitride (hBN) interlayers [54,55] or for molybdenum disulfide (MoSe${}_{2}$) interlayers [56]. Similarly, thin interlayers of In [57] or Se [58] in between the TMD monolayer and the Au metal contacts can lead to atomically sharp van der Waals contacts, even though the specific roles of the In and Se interlayers in this context are different. The In interlayer will directly form the contact with the TMD, while the Au layer will stabilize the 10 nm In layer and prevent its reactions with the environment [57]. The Se interlayer, on the other hand, will evaporate during the post-annealing process. The Se interlayer prevents any damage to the TMD during the Au evaporation, and the contact is then formed directly by the Au layer [58]. Alternatively, Fermi level pinning is strongly reduced for direct 2D/2D contacts formed for example between the semi-metal graphene and 2D semiconductors like MoS${}_{2}$ [59] or tungsten diselenide (WSe${}_{2}$) [60]. In the same manner, the phase engineered metallic 1T phase of a TMD like MoS${}_{2}$ can be used to contact the respective semiconducting 2H phase at a low contact resistance [61]. Another approach aims to lower the contact resistance by narrowing the SB width using local doping. For example, surface charge transfer doping (SCTD) (see Section 5.1) has been used to locally p-dope WSe${}_{2}$ [62] as well as to n-dope MoS${}_{2}$ [63].

### 3.2. Contact Resistances

The smallest contact resistances to 2D semiconductors up to date have recently been reported for semimetallic bismuth contacts to TMDs [27]. Bismuth can suppress metal-induced gap states, thereby creating Ohmic contacts and a small $R_{C}$ of 123 Ω
μm when contacting monolayer MoS${}_{2}$, see Figure 3a. This small contact resistance comes close to the upper acceptable limit according to the IRDS of 111 Ω
μm and is orders of magnitude below most other reports for contacts to monolayer and few layer MoS${}_{2}$, see Figure 3b.

However, in recent works which aimed to reproduce the small $R_{C}$ by Shen et al. [27] for Bi contacts, the obtained contact resistances were considerably higher at 500 Ω
μm [65] or even 2.5 kΩ
μm [30]. Instead, competitively small contact resistances and a higher thermal stability were found for antimony (Sb) contacts with a contact resistance of 659 Ω
μm [65] or down to 146 Ω
μm [30]. Thus, while two promising candidates for Ohmic contacts to n-type semiconductors have been identified with Bi and Sb, there are yet few demonstrations of low-resistive contacts to p-type devices. Recently, good results have been achieved for nitric oxide doped WSe${}_{2}$ FETs with a resistance of 950 Ω
μm [66] or Ru contacts to WSe${}_{2}$ FETs with a resistance of 2.7 kΩ
μm [30].

### 3.3. Nanoscale Contact Lengths

In addition, Shen et al. used in their study top contacts with a contact length ($L_{C}$) of 1 μm [27]. However, for nanoscaled CMOS logic, nanoscaled contacts with $L_{C}=18$ nm are required according to the 0.7 nm node specifications, given by the critical dimensions of the source and drain contacts [4]. As $L_{C}$ is reduced, the contact resistance increases due to current crowding, which can, according to the transmission line model [67], be described by
(1)$$R_{C}=\sqrt{\rho_{C}R_{\mathrm{SH}}}\coth\left(L_{C}/L_{T}\right)$$
with the specific contact resistivity $\rho_{C}$, the channel sheet resistance $R_{\mathrm{SH}}$, and the transfer length $L_{T}=\sqrt{\rho_{C}/R_{\mathrm{SH}}}$. As a small contact resistance requires $L_{C}\gg L_{T}$, the transfer length should be smaller than 10 nm. Also, in this respect, the Bismuth contacts reported by Shen et al. show promise with a transfer length of 8 nm [27], which is orders of magnitude smaller than previous reports at 40 nm [64] and 145 nm [63]. In a double logarithmic graph of $R_{C}$ as a function of $L_{C}$, shown in Figure 3c, $R_{C}$ bends where $L_{C}=L_{T}$. This shows that the contact resistances of the Bi/MoS${}_{2}$ contacts could be reduced for a smaller $R_{\mathrm{SH}}$, hence if the layer quality was improved.

## 4. Gate Stack Design

In order to ensure high performance and reliable operation of 2D FETs, insulating materials which are suitable for the gate stack have to meet numerous requirements. These requirements are categorized into three main areas, good scalability, minimization of insulator-related charge traps and a suitable deposition technology. For nanoscaled devices, a good gate control and a small scale length (λ) are essential which can only be achieved by reducing the equivalent oxide thickness (EOT), given by $\mathrm{EOT}=t_{\mathrm{ins}}\varepsilon_{{\mathrm{SiO}}_{2}}/\varepsilon_{\mathrm{ins}}$. In the IRDS, the EOT targets are specified based on the reported capacitive equivalent thickness (CET) of 0.9 nm, which corresponds to an EOT of about 0.6 nm [4]. At the same time, this EOT reduction makes it more difficult to maintain small gate leakage currents through the gate insulator, as required for small off-state currents and a low power consumption. For low-power applications, the gate leakage current density should be below 10^−2^ A cm^−2^ at an applied gate voltage of up to 0.65 V, as defined by the supply voltage ($V_{\mathrm{DD}}$) [4,68].

### 4.1. Scalable Gate Insulators

Up to now, there have been only few 2D FET prototypes based on a gate stack with an EOT smaller than 4 nm, out of which several are based on the commonly used amorphous insulator hafnium dioxide (HfO${}_{2}$), either as a global back gate for tungsten disulfide (WS${}_{2}$) (EOT=0.7 nm) [42], or in a top-gate layout where HfO${}_{2}$ was deposited with atomic layer deposition (ALD) on MoS${}_{2}$ and WSe${}_{2}$ monolayers (EOT=1 nm) [40]. In addition, the potential of several novel insulators has been explored, including for example crystalline calcium fluoride (CaF${}_{2}$) layers serving as a back gate for MoS${}_{2}$ (EOT=1.2 nm) [69] or the layered native oxide bismuth selenide (Bi${}_{2}$O${}_{5}$Se) to the 2D semiconductor Bi${}_{2}$O${}_{2}$Se (EOT=0.9 nm) [70]. Also, an amorphous antimony trioxide film on MoS${}_{2}$ (EOT=1.6 nm) has been suggested as a scalable gate insulator [71] and the perovskite strontium titanium oxide (SrTiO${}_{3}$) was used as a gate insulator for MoS${}_{2}$ FETs (EOT=1 nm) [41,72]. Unfortunately, the layered, crystalline insulator hexagonal boron nitride (hBN), which is widely considered to be a promising gate insulator for 2D FETs [73,74], is not scalable down to an EOT below 1 nm. Even in the best case scenario, where the impact of charge traps on the gate leakage current via a trap-assisted-tunneling mechanism is neglected, the gate leakage current exceeds the low-power limit for gate voltages above 0.5 V, see the calculated current densities in Figure 4a [75].

### 4.2. Minimization of Insulator-Related Charge Traps

In addition to a good scalability of the gate stack, insulator-related charge traps, including both interface traps as well as border traps within the insulator have to be minimized. Based on the subthreshold slope (SS) which is usually evaluated for 2D FETs, the interface trap density for monolayer MoS${}_{2}$ FETs can be calculated, even though it should be noted that also fast insulator border traps can contribute to the SS. This results in an interface trap density ($D_{\mathrm{it}}$) of about 10^−13^ cm^−2^eV^−1^ at the interface of the amorphous oxides aluminum oxide (Al${}_{2}$O${}_{3}$), silicon dioxide (SiO${}_{2}$) or HfO${}_{2}$ and a considerably smaller interface trap density of 10^−11^ cm^−2^eV^−1^ for hBN/MoS${}_{2}$ FETs [76]. These estimates are in good agreement with the interface trap densities extracted for the interface of MoS${}_{2}$ with amorphous oxides using capacitance-voltage measurements [78] and low frequency noise measurements for hBN/MoS${}_{2}$ FETs [79]. These small interface trap densities are inherently related to the clean van der Waals interface formed between hBN and layered semiconductors [80]. In Figure 4b the van der Waals interface formed between the layered material Bi${}_{2}$O${}_{2}$Se and its native oxide Bi${}_{2}$O${}_{5}$Se is shown in a HAADF TEM image, which indicates a better quality than the interface between transferred thin SrTiO${}_{3}$ on top of few-layers of MoS${}_{2}$, see Figure 4c. This indicates a small density of interface traps at the Bi${}_{2}$O${}_{5}$Se/Bi${}_{2}$O${}_{2}$Se interface, likely comparable to the small interface trap density observed for hBN/MoS${}_{2}$ interface. Furthermore, the insulating environment of the 2D material has a pronounced impact on the mobility of the 2D semiconductor via charged impurity scattering [81] as well as surface optical phonon scattering [82]. In this context, an hBN encapsulation provides the highest mobility for 2D semiconductors according to the current state of knowledge [75,82].

Charge trapping events at border traps in the gate insulator are the root cause for various stability and reliability concerns [83], including low-frequency noise [84], the Bias Temperature Instability (BTI) [85], and the hysteresis in the transfer characteristics [86]. In general, the energetic trap levels of insulator traps form defect bands and their charging time constants span a wide range from nanoseconds up to years [87]. Furthermore, border traps are characterized by a strong gate bias dependence of these charging time constants, as the defect bands are bent by an applied gate voltage [88]. In order to improve the electrical stability of 2D FETs, the probability for charge trapping can be considerably reduced if the defect bands in the gate insulator are energetically far away from the conduction and valence band edges of the 2D semiconductor [77]. For example, n-type WS${}_{2}$/HfO${}_{2}$ FETs are expected to be electrically unstable due to frequent charge trapping events in the electron trapping bands, whereas the stability is likely considerably improved for p-type WS${}_{2}$/HfO${}_{2}$ FETs due to the increased energy barrier for charge trapping, see Figure 4d. By selecting a suitable combination of 2D semiconductor and gate insulator or by tuning this respective alignment with fixed charges at the interface [89] or electric dipoles within the gate stack [87], the number of electrically active charge traps can be minimized. An overview over the band alignment of various 2D semiconductors to the three commonly used amorphous oxides is shown in Figure 4e.

### 4.3. Insulator Deposition Technology

Beyond the concerns for the reduction of charge traps, industrial integration of 2D FETs requires a top-gated device design [28]. Thus, suitable gate insulators have to be deposited uniformly on top of 2D semiconductors. One of the most promising deposition technologies for top-gate integration is ALD, even though the inert basal planes of 2D semiconductors inhibit direct nucleation of the gaseous precursors, which renders the formation of high-quality layers challenging [90]. This problem can be solved either by activating the surface using an, e.g., O${}_{2}$ plasma, even though it was demonstrated that this severely damages the topmost 2D layer [91] or by using a buffer or seeding layer on top of the 2D material [92], where self-assembled polymer monolayers like perylene-3,4,9,10-tetracarboxylic dianhydride (PTCDA) have led to good results [40]. Alternatively, insulators can also be transferred on top of 2D semiconductors [93]. However, while this approach is routinely used for the fabrication of prototypes in university cleanrooms [41,72], its scalability remains questionable. Finally, the in-situ transformation of a layered semiconductor into its native oxide [70], or the direct evaporation of an amorphous insulator on top of a 2D semiconductor [71], are promising routes for top-gate integration.

## 5. CMOS Process Integration

In addition to the aforementioned challenges, successful CMOS integration of 2D materials requires the realization of n-type and p-type FETs next to each other. This, in turn, is linked to suitable stable doping schemes for 2D monolayers, compatible 2D semiconductors for p-type and n-type operation, and the possibility to tune the threshold voltage ($V_{\mathrm{th}}$) for example by using the metal gate work function.

### 5.1. Doping 2D Semiconductors

In general, substitutional doping in 2D materials has been realized by replacing both the cationic components (e.g., replace Mo with Nb in MoS${}_{2}$) [94] and the anionic components (e.g., replace S with N in WS${}_{2}$) [66,95] of TMDs [48]. However, substitutional doping schemes in 2D materials typically lead to severe lattice defects in the 2D layer and poor dopant activation despite structural incorporation of dopants [28]. In contrast, a more effective doping scheme is surface charge transfer doping (SCTD) [62,63], where interfacial charges determine the doping of the 2D semiconductor. These interfacial charges are introduced either via a sub-stoichiometric oxide layer [63,96] or oxygen plasma treatment [97]. Nevertheless, these additional interfacial charges also tend to degrade the mobility and subthreshold slope of the FETs as well as their electrical stability, in particular for monolayer FETs [62]. In fact, also for SCTD, charged defects are introduced directly at the surface of the 2D channel, defects that lead to pronounced charge impurity scattering at the ionized dopants.

This problem can be addressed with the concept of modulation doping, where the charge dopants are spatially separated from the conduction channel. Recently, this concept of remote modulation doping for 2D TMD channels has been theoretically proposed [98] and experimentally demonstrated [99,100]. In these works, a MoS${}_{2}$ channel is separated from the molecular dopants using a few layers of hBN as a tunnel barrier, effectively suppressing charged impurity scattering and achieving a sizable mobility enhancement [99]. The main disadvantage of this doping concept is that the additional tunnel barrier and doping layer increases the EOT of the gate stack, thereby reducing the gate control. Thus, which doping method is best suited for ultra-scaled 2D devices is currently unclear.

### 5.2. 2D CMOS Inverters

Integrated CMOS circuits require n-type and p-type FETs to be processed in the direct vicinity of each other. The most straightforward way of testing this possibility for very-large scale integration (VLSI) is to fabricate CMOS inverters using the previously discussed doping schemes. At the same time, CMOS inverters are important benchmark circuits for the performance, variability, and electrical stability of digital logic [101]. Up to now, CMOS inverters based on 2D materials have often combined different TMDs such as MoS${}_{2}$ for the n-type FET and WSe${}_{2}$ for the p-type FET, even though this method makes the integration of n- and p-type FETs in close vicinity as well as the tuning of $V_{\mathrm{th}}$ difficult [102,103]. As an alternative, one 2D semiconductor has been doped either substitutionally [104], electrostatically with an additional gate [105,106] or by selecting different metal contacts for n- and p-type [106,107]. For example, black phosphorous (BP) FETs have been fabricated with Cr/Au contacts for the p-type FET and Ti contacts for the n-type FET, see Figure 5a. Recently, van der Waals contacts from the back side have been formed to create n-type WSe${}_{2}$ FETs with In contacts and p-type WSe${}_{2}$ FETs with Pt contacts [108]. In this way, voltage gains of up to 198 for $V_{\mathrm{DD}}=4.5$ V [108], 80 for $V_{\mathrm{DD}}=2$ V [103] and up to 20 for $V_{\mathrm{DD}}=0.4$ V [106] have been achieved, respectively. The gains on BP inverters at $V_{\mathrm{DD}}=0.4$ V and 0.6 V, see Figure 5b,c, are compatible with a scaled $V_{\mathrm{DD}}$ in scaled technology nodes at $V_{\mathrm{DD}}\leq 0.65$ V [4].

### 5.3. Wafer-Scale Industrial Processing

Additionally, successful CMOS integration requires processing steps which allow for a high yield, high process uniformity, high reproducibility and low variability on a large scale in an industrial process flow. Recently, considerable progress has been made in this regard in the research departments of semiconductor industry, e.g., at Intel [110], TSMC [111] and IMEC [112]. Processes for the deposition of 2D materials using metal-organic chemical vapor deposition (MOCVD) have been developed on 300 mm wafers for MoS${}_{2}$ as a channel material for n-type FETs and WSe${}_{2}$ as a channel material for p-type FETs [30,113] and WS${}_{2}$ exhibiting ambipolar behavior where the polarity is determined by the back gate [37,109], see Figure 5d. As an example of a 300 mm wafer-scale process flow for 2D semiconductor-based FETs, the fab integration route of WS${}_{2}$ FETs, as developed by imec [109,112], is described in the following. In the first step, the WS${}_{2}$ layer is directly grown on the back gate stack consisting of HfO${}_{2}$ and SiO${}_{2}$ using an MOCVD growth process. A high-temperature MOCVD growth at 750 °C or 950 °C provides a better layer quality than a low-temperature ALD growth process, with larger grain sizes and a higher degree of crystallinity. Nevertheless, the layer quality limits the effective electron mobility to about 0.5 cm^2^/Vs, which is considerably worse than that in exfoliated flakes [112]. As a next step, a Si seed deposited with a molecular beam serves as a nucleation site for the ALD growth of the HfO${}_{2}$ top gate oxide. This weak adhesion between the layered WS${}_{2}$ and the surrounding oxides is a critical problem for conventional patterning steps based on a SiO${}_{2}$ hard mask. Instead, a spin-on process for a soft etch mask was developed to reduce the mechanical stress on the layered WS${}_{2}$ in order to avoid local delamination [114]. In the next step, a damascene side contact etch is performed to form Ti side contacts at source and drain. Currently, side contacts are the only possible contact geometry, as no selective etching process is available, which would stop etching at the thin WS${}_{2}$ layer [109]. Next, a damascene top gate process is used to create the top gate contact, and in the last step, vias are formed, leading to the final device geometry [114], shown in the TEM cross section in Figure 5d.

In addition, small contact resistances have been achieved for semimetallic antimony (Sb) contacts for n-type MoS${}_{2}$ FETs [65,115] and ruthenium (Ru) contacts for p-type WSe${}_{2}$ FETs [30], aiming for a Schottky depinning by a reduction of metal induced gap states [27]. Nevertheless, at the moment in particular the contact resistance for p-type TMD FETs is yet one order of magnitude too high to be competitive to Si-technology [30,115]. Furthermore, a transfer scheme for MOCVD grown TMD films has been developed using a polymer on glass [109] as shown in detail in Figure 5e. Alternatively, weakly bonded Bi has been used as a contact layer for the second sacrificial transfer wafer [111]. A key advantage of such a transfer step is that the best quality TMDs are typically grown at high temperatures above 900 °C, while an integration at the back-end of the line (BEOL) only allows for a thermal budget of up to 400 °C. Therefore, a transfer step is required for BEOL integration of TMD films and the first demonstrations show promise even though homogeneity and success rate over the 300 mm wafer yet needs to be improved [109,111], see Figure 5e.

## 6. Conclusions

In recent years, there has been tremendous progress in the area of 2D FETs. For example, double-gated WS${}_{2}$ FETs with 2 nm gate length have been fabricated on 300 mm wafers [37], record-low contact resistances have been achieved with Bi contacts to MoS${}_{2}$ [27], and a variety of gate insulators which form a van der Waals interface with 2D materials have been discovered. At the same time, the challenges which need to be addressed for the integration of 2D materials at the FEOL in order to extend the scaling regime to gate lengths of below 12 nm are numerous. First of all, stacked 2D FETs with scaled channel lengths, widths, contact lengths, and gate pitches are currently far from being realized. Stacked designs require either a reduction of the MOCVD growth temperature for high quality 2D films or a homogeneous transfer process which can deliver the necessary yield on an industrial scale [109]. In addition, despite low contact resistances having been achieved for n-type MoS${}_{2}$ FETs, contact resistances for p-type FETs still exceed the target value by one order of magnitude [66] and scaled contact lengths are challenging. Finally, one of the major obstacles for 2D CMOS is the identification of a suitable gate insulator. In fact, a semiconductor/insulator system has to be identified with a clean van der Waals interface which provides small remote charge carrier scattering and thus high mobilities in the semiconductor and at the same time good scalability and few electrically active border traps in the insulator [76]. Despite the proposal of concepts like Fermi-level tuning to enhance the electrical stability when using amorphous oxides, it remains unclear whether amorphous oxides in combination with 2D semiconductors will ever be able to meet the requirements in terms of variability and reliability. More promising insulator/semiconductor systems, like Bi${}_{2}$O${}_{5}$Se/Bi${}_{2}$O${}_{2}$Se [70], CaF${}_{2}$/MoS${}_{2}$ [69], or SrTiO${}_{3}$/MoS${}_{2}$ [72] have been suggested, but for all of these systems, there is a wealth of questions unanswered so far.

Henceforth, the future role 2D materials might play for nanoscaled CMOS logic remains unclear. Their commercial success and adaptation will likely depend on whether, over the next decade, a “killer application” will be found that acts as the first enabler of commercial microchips, including 2D materials. Most likely, only the demonstration of one successful and profitable use case will help to attract the large volume of research funding required to tackle all the remaining challenges outlined above. These solutions will pave the way for many other application scenarios and eventually 2D materials as channels for ultra-scaled FETs at the FEOL. Such possible “killer applications” for 2D materials could, for example, be the monolithic 3D integration of 2D-based devices in the BEOL or flexible electronics based on 2D materials. In any case, we believe that in view of the short history of 2D transistors, the lab-to-fab transition of 2D materials is just beginning. Over the course of the next two decades, we will likely reach a stage where 2D materials can bring measurable benefits to our society.

## Figures and Tables

**Figure 1 nanomaterials-12-03548-f001:**
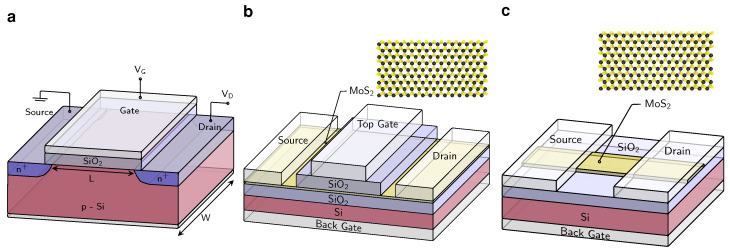
(**a**) Schematic drawing of a planar n-type silicon MOSFET. (**b**) Schematic layout of a large area FET with an MoS${}_{2}$ channel. Here, SiO${}_{2}$ is the back gate oxide with Si serving as the global back gate and the top gate oxide with a local top gate contact. This double-gated configuration provides the best gate control. In the upper right corner, the atomic structure of a monolayer of the semiconducting 2H phase of MoS${}_{2}$ is shown. The black spheres represent Mo atoms and the yellow spheres S atoms. (**c**) Layout of a small area back-gated MoS${}_{2}$/SiO${}_{2}$ FET.

**Figure 2 nanomaterials-12-03548-f002:**
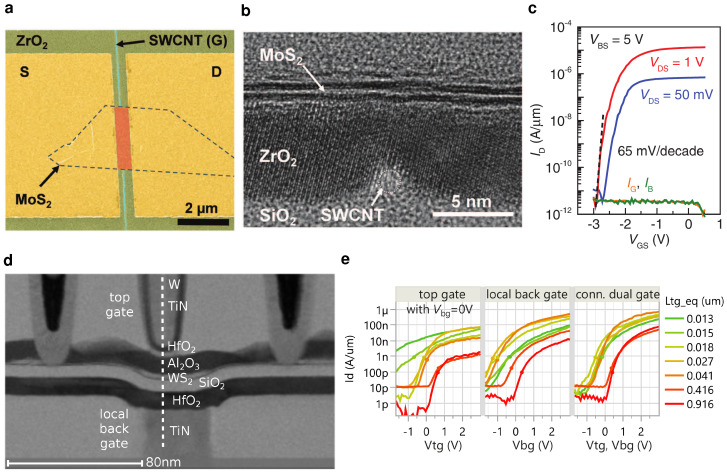
(**a**) False–colored scanning electron microscopy image of a MoS${}_{2}$ FET with a channel length of 1 nm. The MoS${}_{2}$ flake (in orange) is gated by a single walled carbon nanotube (SWCNT, in cyan), contacted by Ni source and drain electrodes (in yellow), using ZrO${}_{2}$ as a gate insulator (in green). (**b**) Cross-sectional TEM of the MoS${}_{2}$/ZrO${}_{2}$/SWCNT transistor, where the SWCNT gate, the ZrO${}_{2}$ insulator and the layered MoS${}_{2}$ channel can be seen. (**c**) $I_{D}-V_{\mathrm{GS}}$ characteristics of the MoS${}_{2}$/ZrO${}_{2}$/SWCNT FET at a bias of $V_{\mathrm{BS}}=5$ V, which was applied to the global silicon back gate to n-dope the access regions. A steep subthreshold slope of 65 mV/dec. is achieved. (**a**–**c**) reproduced with permission from S. Desai et al., Science 354, 96–99, 2016; published by the American Association for the Advancement of Science [34]. (**d**) Cross-sectional TEM of a double-gated WS${}_{2}$ transistor using few-layer MOCVD deposited WS${}_{2}$ as a channel and ALD grown HfO${}_{2}$ as a gate insulator. To the right of the dashed line the materials of the two gate stacks are listed: W-TiN-HfO${}_{2}$-Al${}_{2}$O${}_{3}$-WS${}_{2}$-SiO${}_{2}$-HfO${}_{2}$-TiN-n${}^{+}$Si. In the connected dual gate design, the back gate is connected via the n${}^{+}$ well to the top gate. (**e**) Comparison of the $I_{D}-V_{\mathrm{GS}}$ characteristics of the HfO${}_{2}$/WS${}_{2}$/HfO${}_{2}$ FET for either a top-gated, back-gated or double-gated configuration of varying top gate lengths. Short channel effects are best suppressed in the double-gated design. (**d**,**e**) reproduced with permission from Q. Smets et al., IEDM, 725–728, 2021; published by the IEEE [37].

**Figure 3 nanomaterials-12-03548-f003:**
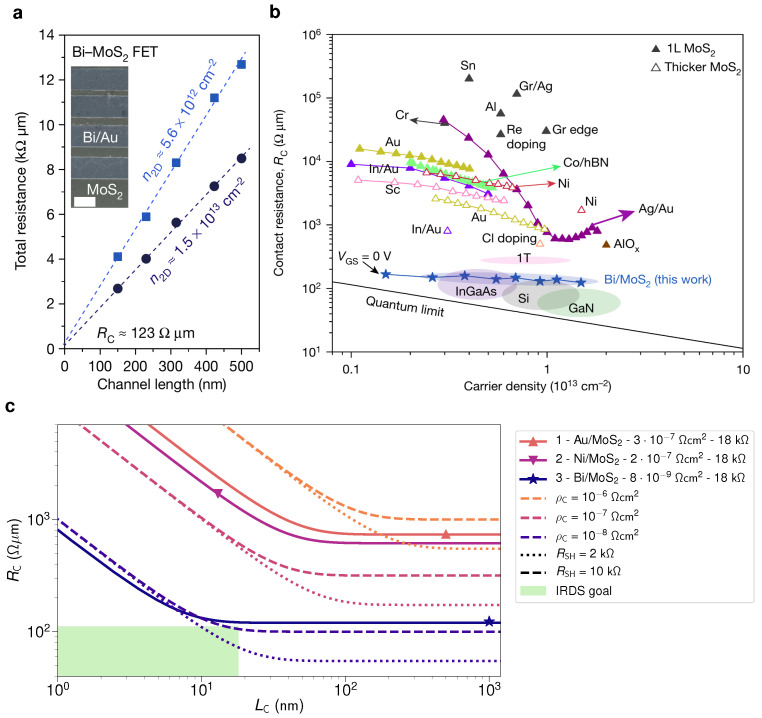
(**a**) Transfer length measurements (TLM) for Bi contacts to a monolayer of MoS${}_{2}$ using a 100 nm thick SiN${}_{x}$ gate insulator. The total resistance is measured for different channel lengths at different gate overdrive voltages, corresponding to different carrier densities, $n_{2}D$. The intercepts of the linear regression through the measured resistances with the y-axis at L=0 nm determine the contact resistance of 123 Ω
μm. In the inset, a false-color SEM image of the TLM structure is shown (scale bar 1 μm). (**b**) Comparison of contact resistances $R_{C}$ reported in the literature for monolayer and multilayer MoS${}_{2}$ as well as conventional semiconductors InGaAs, Si and GaN as a function of the carrier density $n_{2}D$. A solid black line indicates the quantum limit of $R_{C}$, which is determined by the quantum resistance $h/2q^{2}$ and the number of conducting modes per channel width, which is related to $n_{2}D$. (**a**,**b**) reproduced with permission from P. Shen et al., Nature 593, 211–217, 2021; published by Springer Nature [27]. (**c**) Calculated $R_{C}$ as a function of the contact length $L_{C}$ for several combinations of the specific resistivity $\rho_{C}$ and the sheet resistance $R_{\mathrm{SH}}$. At the bottom left corner, the green area indicates the targeted regime [4]. In comparison, solid lines show reported values from references 1 [64], 2 [33] and 3 [27]. The Bi contacts to MoS${}_{2}$ come close to the target, even though the sheet resistance of the MOCVD grown monolayer MoS${}_{2}$ is too high.

**Figure 4 nanomaterials-12-03548-f004:**
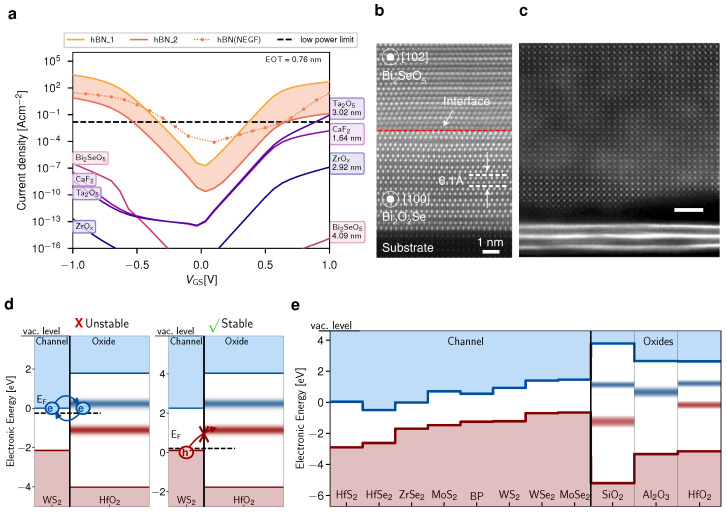
(**a**) Calculated gate leakage current densities through a MOS structure formed by a gold metal contact, various gate insulators, and a silicon channel. The currents are compared as a function of the applied gate voltage for a scaled EOT of 0.76 nm and were calculated based on the Tsu-Esaki model. The orange range indicates the interval of possible currents through hBN spanned by two different sets of effective masses and the dotted orange line shows the DFT+NEGF results. The leakage currents through hBN exceed the leakage currents through native oxides or ionic insulators by many orders of magnitude. Reproduced with permission from T. Knobloch et al., IRPS, 2A.1, 2022; published by the IEEE [76]. (**b**) Cross-sectional high-angle annular dark-field (HAADF) TEM image of the interface between the semiconductor Bi${}_{2}$O${}_{2}$Se and its native oxide Bi${}_{2}$O${}_{5}$Se, after few minutes of oxidation at 380 °C. An atomically sharp Van der Waals interface between both compounds is formed during oxidation. Reproduced with permission from T. Li et al., Nature Electronics 3, 473-478, 2020; published by Springer Nature [70]. (**c**) HAADF TEM image of the interface formed when transferring a thin crystalline SrTiO${}_{3}$ on top of few-layers of MoS${}_{2}$, scale bar 2 nm. Reproduced from A. Yang et al., Nature Electronics 5, 233–240, 2022e [72]. (**d**) If the conduction band edge of the 2D semiconductor is close to a charge trapping band in the oxide, charge trapping events are frequent and the FET is electrically unstable. This is shown in the band diagram of n-type WS${}_{2}$ and its alignment to the electron trapping band in HfO${}_{2}$. For p-type WS${}_{2}$, however, the FET will be more stable. (**e**) Band diagram of various 2D semiconductors with the three most commonly used amorphous oxides, indicating good stability, for example for black phosphorous (BP) or hafnium disulfide (HfS${}_{2}$) FETs with HfO${}_{2}$ as a gate insulator. (**d**,**e**) reproduced from T. Knobloch et al., Nature Electronics 5, 356–366, 2022 [77].

**Figure 5 nanomaterials-12-03548-f005:**
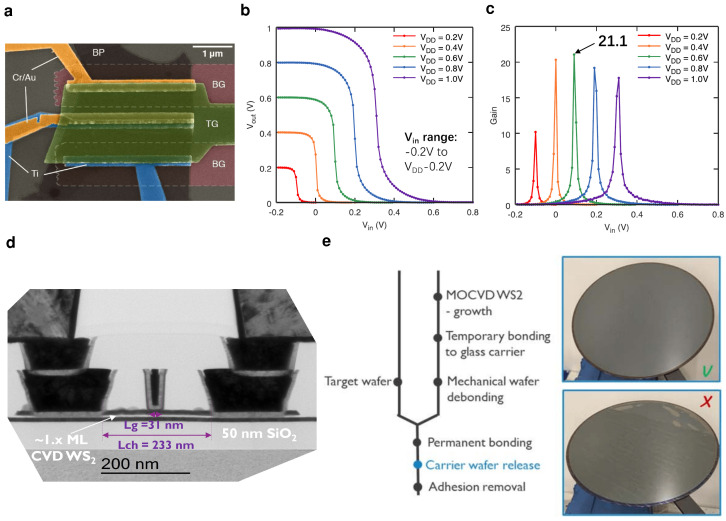
(**a**) False–colored SEM image of black phosphorus (BP) FETs with Cr/Au contacts for the p-type FET (top) and Ti contacts for the n-type FET (bottom). A 4 nm thick Al${}_{2}$O${}_{3}$ layer serves as both the top gate (TG) and the back gate (BG) gate insulator and the two separate BGs are used to adjust the threshold voltages of p- and n-type FET. These two FETs form an inverter with the input connected to the TG and the output to the shorted Ti and Au contacts at the center. (**b**) Voltage transfer characteristics of the BP/Al${}_{2}$O${}_{3}$ inverter with local back gates at different supply voltages $V_{\mathrm{DD}}$. (**c**) Gain of the BP/Al${}_{2}$O${}_{3}$ inverter at different $V_{\mathrm{DD}}$, with the highest gain of 21.1 observed for $V_{\mathrm{DD}}=0.6$ V. (**a**–**c**) reproduced with permission from P. Wu et al., DRC 2018; published by the IEEE [106]. (**d**) TEM image of double-gated WS${}_{2}$ FETs fabricated in a 300 mm pilot line, which use HfO${}_{2}$ as a top gate and a HfO${}_{2}$/SiO${}_{2}$ stack as a back gate oxide. The device has a short top gate length of 31 nm, a total channel length of 233 nm and trench side contacts to the few-layer WS${}_{2}$ channel. Reproduced with permission from Z. Ahmed et al., IEDM 2020, published by the IEEE [25]. (**e**) Schematic of the transfer process for WS${}_{2}$ grown on a 300 mm wafer. The MOCVD growth process is performed at a high temperature of above 900 °C, then the wafer is bonded to a glass carrier, mechanically debonded from the growth substrate before it is permanently bonded to the target wafer. On the right side, images show that the transfer can be uniform (top) and non-uniform (bottom) indicating a need for further process improvement. Reproduced with permission from I. Asselberghs et al., IEDM 2020; published by the IEEE [109].

## Data Availability

Not applicable.

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
