# Peer review of "Challenges for Nanoscale CMOS Logic Based on Two-Dimensional Materials"

_nanomaterials, 2022, doi:10.3390/nano12203548_

Round 1

Reviewer 1 Report

In this review, Theresia Knonloch et al. discuss the most important problems that 2D FETs are currently facing and provide an overview about their limitations as well as pointing towards the most important future research questions. This is a timely review on a very important question: how to further scale down the size of transistors and improve the performance of electronic chips. And this review has a clear structure and a comprehensive overview in this area. However, to give a better understanding to readers, I recommend the authors add necessary schematic diagrams to give some intuitive feelings. For example, the differences between 2D CMOS logical circuits and conventional CMOS, an imagined structures of electronic chips based on 2D materials, or a timeline to conclude the history and progress of this area.

Therefore, I recommend acceptation of this manuscript after the mentioned improvement.

Reviewer 2 Report

The manuscript presents a review on CMOS logic circuit to be used in 2D materials. The subject is extremely current and it has strong implications on different applicative fields. The review is well organized and the number and type of references are appropriate. Although some parts can be devoted to the specialists of the field, it provides a picture  of the state of art and then it can be useful also for generic readers.   

My observation concerns the Conclusions  Section which should be adjusted by providing, besides the summary of the results, some more conclusive remarks.

Reviewer 3 Report

The manuscript reviews the recent achievement and challenges of 2D material-based CMOS logic devices for ultra-scaled technology nodes. It mainly discussed the four critical challenges, device scaling, contact issue, gate design, and wafer scale integration, impeding the 2D material logic device application. This review covered recent research in related fields and proposed several valuable ideas. Here, I have several comments to put the manuscript forward.

1. The title of the paper seems too large compared to the content described in the manuscript. The author mainly focused on four points, scaling, contact issues, gate design, and wafer scale processing, which focus more on the 2D material-based CMOS logic in commercialized applications. The author may highlight the main point of the paper by changing the title to a more proper one.

2. In line 78, the author claimed that the smallest gate length based on 2D FET is 1 nm, fabricated with the carbon nanotube. But as far as I know, one paper [Nature 603,  259–264 (2022)] reported the graphene gate with sub-1nm length. Please refer to this paper and revise the paragraph in section 2.1 and Figure 1.

3. In section 2.2, it would be better if the author points out the main reason for the difficulty in the ultra-scaled engineering(means sub 10 nm) of 2D FET compared to the Si from the material property point of view.

4. In line 155, the author discusses that the Se in between TMD and Au metal contact can lead to the formation of atomically sharp van der Waals contact. Readers may confuse because the Se does not function as the contact in that paper. The Se will be evaporated during the post-annealing process from the interface. The function of Se is to prevent the damage of MoS2 crystal by the Au atoms in the deposition process. Please write more specifically about the role of Se here.

5. In section 5.1, the author only described substitutional doping and charge transfer doping. There may be more doping mechanisms, such as remote modulation doping [Nature Electronics 4, 664–670 (2021)]. Please do a more paper search on the 2D material doping and describe it in detail.

6. In section 5.2, the author describes the recent result of the CMOS inverter. However, I could not find why the author discusses the inverter in this part. In section 5, the author tends to discuss wafer scale processing. Please consider whether this section is necessary here and if so, please describe the reason.

7. In section 5.3., there has a significant part describing the contact issues. The author should focus more on industrial processing than contact resistance, as discussed in the previous section of this manuscript.

Reviewer 4 Report

The authors review quite well efforts for recent designing CMOS circuits with 2D structures, mainly of MoS2 and WS2. This review complements another one published recently by the same group in Advanced Materials (https://doi.org/10.1002/adma.202201082). The topic is appropriate for Nanomaterials, however the authors should account that this journal is primarily about the materials. Therefore, to properly meet the journal’s audience and enhance the readers’ interest (in terms of authors and journal interests), it is advised:

1) to introduce the section, next to Introduction, of the overview the structures of the available 2D materials which are further considered in sections 2-5;

2) while the title says about “2D materials” the authors mostly consider Transition Metal Dichalcogenides and nothing say about other structures, like Mxenes or graphene ones. So, the authors have to state in Introduction why do they limit their review by TMDs or to change the title via introducing “TMD” there. Otherwise, they have to extend the review and account for other publications like ACS Nano 15(2021) 19266, Small 17(2021) 2101482, etc (if to talk about Mxenes); there are much more works for employing the graphene.

Reviewer 5 Report

In the review, the authors report on the current status of 2D FETs for the implementation of CMOS logic complying with the IRDS. The authors motivate the need for 2D semiconductors referring to the planned node size reduction and present existing academic prototypes as well as the fabrication methods discussing the scalability. In the subsequent sections, the concepts for contact fabrication and gate design are outlined including a discussion of the influence of oxide and interface traps on the transfer characteristics and gate current. Finally, the potential for CMOS process integration is discussed.

The article is well written and provides a good overview on the current status of 2D FETs based on transition metal dichalcogenides. The discussion focuses on materials and design and discusses the related semiconductor physics particularly for the gate design. The authors might consider discussing the transient and thermal characteristics, but this is optional.

In line 225 (on page 8) the subthreshold slope is attributed to the interface traps but also the oxide traps contribute. The authors might want to point his out here. In the subsequent sections oxide traps are discussed. In this context. The authors might want to point out that the interaction of oxide traps and the CB/VB in the semiconductor is subject to the gate-source voltage.

In line 160 the abbreviation TMD is introduced but has been used before in line 155.

The authors might want to introduce the abbreviation BP also in the caption of Fig. 3.

Publication can be recommended if the minor issues are addressed.

Round 2

Reviewer 4 Report

The authors have properly revised the review regarding the comments.